# Observation-Free Attacks on Stochastic Bandits

**Yinglun Xu**
University of Illinois at Urbana-Champaign
`yinglun6@illinois.edu`

**Kumar Bhuvesh**
Georgia Institute of Technology
`bhuvesh@gatech.edu`

**Jacob Abernethy**
Georgia Institute of Technology
`prof@gatech.edu`

## Abstract

We study data corruption attacks on stochastic multi arm bandit algorithms. Existing attack methodologies assume that the attacker can observe the multi arm bandit algorithm's realized behavior which is in contrast to the adversaries modeled in the robust multi arm bandit algorithms literature. To the best of our knowledge, we develop the first data corruption attack on stochastic multi arm bandit algorithms which works without observing the algorithm's realized behavior. Through this attack, we also discover a sufficient condition for a stochastic multi arm bandit algorithm to be susceptible to adversarial data corruptions. We show that any bandit algorithm that makes decisions just using the empirical mean reward, and the number of times that arm has been pulled in the past can suffer from linear regret under data corruption attacks. We further show that various popular stochastic multi arm bandit algorithms such UCB, $\epsilon$-greedy and Thompson Sampling satisfy this sufficient condition and are thus prone to data corruption attacks. We further analyse the behaviour of our attack for these algorithms and show that using only $o(T)$ corruptions, our attack can force these algorithms to select a potentially non-optimal target arm preferred by the attacker for all but $o(T)$ rounds.

## 1 Introduction

Multi-armed bandit problems provide a foundational framework for understanding sequential decision making. In the classical setting, on each round of the decision process a learner selects an action (arm) from various alternatives and, upon making this choice, receives some scalar-valued feedback/reward for the chosen action but no additional information. Algorithms for such multi-armed bandits have been widely adopted in various applications, including recommender systems Bouneffouf et al. [2012], Li et al. [2011a], Kawale et al. [2015], Li et al. [2011b] and in numerous modern industry and business applications Villar et al. [2015], Schwartz et al. [2017].

A frequent model assumption for bandit problems is that the reward associated with an arm is a stochastic quantity drawn from fixed distribution associated with each arm, and that this random variable is independent of the learner's previous actions. An alternative assumption, which takes a worst-case perspective and has also been widely studied, is that on every round the reward released by each arm is instead chosen by an adversary which may aim to hurt the learner's learning objective.

The stochastic model is often criticized for being unrealistic: data collected in a sequence rarely satisfy the IID assumption, and it would be naïve to think that corruptions never occur. The adversarial model, on the other hand, is considered highly pessimistic in contexts where we expect learning to be reasonably possible. Researchers have begun to consider intermediate model assumptions, where the input data is generally assumed to be stochastic for the most part, yet a small fraction of malicious

corruptions will occur. One does not have to look hard to find pertinent examples, e.g. click fraud in online advertising [Haddadi, 2010], and fake reviews in online recommendation systems [Wilbur and Zhu, 2009, Kshetri, 2010, Lappas, 2012, Lappas et al., 2016] to name a few.

Understanding adversarial attacks against machine learning algorithms is critical for helping to design robust systems that can be deployed in the wild. There is a long line of work on understanding adversarial data-poisoning attacks against deep learning algorithms [Madry et al., 2017, Akhtar and Mian, 2018, Yuan et al., 2019], supervised learning algorithms [Dai et al., 2018, Liu et al., 2019], and more recently for multi-armed bandit problems . Perhaps the most popular algorithm for the stochastic multi-armed bandit setting, UCB [Auer et al., 2002], has a tight theoretical guarantee on its performance (i.e. its *regret*). Despite all this, it has been shown indeed that UCB is highly vulnerable to data corruption attacks [Jun et al., 2018, Liu and Shroff, 2019]. In short, with only a handful of corruptions on the reward feedback given to the learning, UCB can be tricked into directing most of its choices onto a sub-optimal arm. Adversarial corruptions for multi arm bandit strategies have been studied across two axes: one line of work focus on designing and analysing different techniques to attack existing bandit algorithms [Jun et al., 2018, Liu and Shroff, 2019, Ma et al., 2018, Garcelon et al., 2020], while the other focuses on designing robust algorithms that can perform well under various levels of data corruption [Lykouris et al., 2018, Gupta et al., 2019, Kapoor et al., 2019].

Notwithstanding these prior lines of work, there remains a major gap in the corruption models considered for such adversarial attacks on bandit algorithms. Most existing results assume that the adversary (corruption agent) is given full knowledge of the arm chosen by the learner and can perform a targeted corruption on just the reward selected by the algorithm. It has indeed been shown that all no-regret stochastic bandit algorithms are vulnerable to such powerful adversaries [Liu and Shroff, 2019]. On the other hand, the development of robust algorithms (e.g. Lykouris et al. [2018], Gupta et al. [2019]) have obtained guarantees only under a *weaker* adversary, one that can only corrupt the reward feedback *before* observing the arm selected by the learner. There has been no work, to our knowledge, that has tried to design adversarial attacks against popular stochastic bandit algorithms under the weaker adversary. For algorithms that are deterministic, which select each arm via a non-random function of prior observations, there is no relevant distinction between the strong and weak adversarial models. But given that randomization is a common and important tool in algorithm design, in this work we consider attacks against both randomized and non-randomized algorithms.

With this in mind, the goal of the present paper is to design a method of adversarial attack which (a) is effective against a very broad range of multi-armed bandit algorithms and (b) fits within the weaker adversary model. More specifically, we show that if a stochastic bandit algorithm makes its decisions as a function of a natural statistic, the empirical mean reward and the number of pulls of each arm, then such an algorithms is fully vulnerable to the corruption attacks. This family of bandit algorithms is indeed quite broad, and we show that most of the popular classical strategies—UCB, $\epsilon$-greedy, and Thompson sampling [Agrawal and Goyal, 2012], all of which we analyze—fall within this framework and are thus similarly vulnerable. We further show that using by corrupting only $o(T)$ rounds, our attack can force these algorithms to select a specific arm preferred by the adversary (target arm) for all but $o(T)$ rounds. We believe this reveals what is a core flaw inherent in many bandit algorithms, and these insights can thus help to design more robust learning algorithms in this and other settings.

Compared to the most related works of Jun et al. [2018], Liu and Shroff [2019], Garcelon et al. [2020] which also study adversarial attacks against bandit algorithms, there are three fundamental differences. The first difference is that this line of work assumes that the adversary can observe the actions of the bandit algorithms. This allows the adversary to attack the algorithms based on whether a particular arm is selected or not. Without such ability, to simulate their attack, the adversary need to corrupt all rounds if the bandit algorithm is randomized. The second difference is the corruption model. In their model, the corruption is counted only for the arm which is selected, while in our model, if in a round an arm is corrupted but not selected by the bandit algorithm, we still count it as a corrupted rounds. Based on our notion of corruption, the corruption budget is $T$ for the attackers who need to corruption every round even if most of its corruption is not observed by the algorithm. The third difference is that the attacks in this line of works never apply corruptions on the target arm. Although this makes the target arm more preferred by the bandit algorithm, the negative side effect is that the amount of corruption they can apply on the non target arms is limited because they are picked less often, and thus they may have to keep attacking the algorithm. In our attack, the adversary corrupts all arms at the beginning, making all arms look similar, thus even non-target arms are picked often enough in the early phase of the attack. This allows the adversary to apply enough corruptions

on the non-target arms so that the estimates cannot recover even after the attack stops. Through this attack, we show that all mean based algorithms which make decisions only based on estimates of empirical means are vulnerable to adversarial data corruption attacks. Liu and Shroff [2019] provide a similar conclusion for the offline setting by analysing a few specific algorithms. Also note that in the offline setting considered in Liu and Shroff [2019], the algorithm receives a batch of data with size $T$ at once, and goal of the adversary is to manipulate the algorithm's choice at the $T + 1$ round, which is very different to the online setting.

## 2 Preliminaries

Let's begin by recalling the stochastic multi arm bandit setting. A principal (or learner) faces a sequential decision making problem where it needs to select one out of $K$ actions or arms at each of the $T$ rounds. The principal gets a reward in each round based on the arm chosen in that round. Formally, at each round $t$, the environment generates a reward vector $\boldsymbol{r}^t = (r_1^t, \ldots, r_K^t)$ (not observable to the principal) where $r_i^t \in [0, 1]$ is the reward the principal will receive if arm $i$ is picked, and for each arm $i$, $r_i^t$ sampled from a fixed arm dependant distribution with mean $\mu_i$ which is unknown to the principal. Let $\boldsymbol{\mu} = (\mu_1, \ldots, \mu_K)$ be the mean reward vector that includes mean rewards of all arms. The principal then selects an arm $I^t$ and receives the corresponding reward $r_{I^t}^t$ and does not observe the rest of the values in $\boldsymbol{r}^t$. To characterize the performance of a bandit algorithm, the notion of regret is introduced. The regret of a bandit algorithm is defined as the gap between the total expected reward of the algorithm and the expected reward of the algorithm that always selects the arm with the highest mean reward in each round.

**Definition 1** (Regret).

$$R(T) = T \cdot \max_i \mu_i - \sum_{t=1}^{T} \mu_{I^t}$$

*where $I^t$ is the arm chosen by the algorithm in round $t$.*

Let arm $i^*$ be the optimal arm, i.e. $i^* = \operatorname{argmax}_i \mu_i$. Next we introduce the notion of adversarial attacks in the stochastic bandit setting. The adversarial attack is a form of data corruption where a malicious agent intends to manipulate the behavior of the bandit algorithm by corrupting the reward vector $\boldsymbol{r}^t$ generated by the environment. Specifically, the adversary can change the reward vector $\boldsymbol{r}^t$ to another corrupted reward vector $\hat{\boldsymbol{r}}^t = (\hat{r}_1^t, \ldots, \hat{r}_K^t)$ such that $\hat{r}_i^t \in [0, 1]$ for all $i$. We say that the round $t$ is corrupted if the adversary changes the reward for at least one of the arms, i.e. $\|\boldsymbol{r}^t - \hat{\boldsymbol{r}}^t\|_1 > 0$. Let $C$ be the total number of rounds that the adversary corrupts, that is $C = \sum_{t=1}^{T} \mathbb{1}\{\|\boldsymbol{r}^t - \hat{\boldsymbol{r}}^t\|_1 > 0\}$. We call $C$ the corruption level of the bandit algorithm. Importantly, We assume that the adversary corrupts the reward without observing the arm selected by the adversary. Formally, the protocol between the learner and the adversary at each round $t = 1, \ldots, T$ is as follows:

1. The learner decides a distribution $\pi^t \in \Delta_K$ over $K$ arms.
2. The environment generates a stochastic reward $\boldsymbol{r}^t$.
3. The adversary corrupts the reward, and the corrupted reward becomes $\hat{\boldsymbol{r}}^t$
4. The learner picks an arm $I^t$ from the distribution $\pi^t$ and receives corrupted reward $\hat{r}_{I^t}^t$

Next we give definitions to measure the robustness of an algorithm against adversarial data corruption attacks and the power of attack methods. To characterize the performance of an algorithm under any possible adversarial attack, we introduce the definition of vulnerable algorithms.

**Definition 2** (Vulnerable bandit algorithms). *We say a bandit algorithm is vulnerable if there exists an instance and an adversary such that the adversary with $C = o(T)$ corruption level can induce linear regret $R(T) = \Omega(T)$ on the bandit algorithm in expectation.*

To characterize the performance of an adversarial attack, we need to consider the bandit algorithm it attempts to attack as well. The adversarial attacks that we consider in this work have a goal which is one step harder than just making the bandit algorithm obtain linear regret. The adversary has a favorite arm (that we call the target arm) and the adversary's goal is ensure that the bandit algorithm selects the target arm for most of the rounds of the algorithm. We say a bandit algorithm $\mathcal{B}$ is *completely vulnerable* to an adversarial attack $\mathcal{A}$, if with probability at least $1 - \delta(T)$, with $\delta(T) = o(1)$, the

adversary can make the algorithm pick the target arm specified by the adversary for all but $o(T)$ rounds by using only $C = o(T)$ corruption level.

We now introduce a framework that is typically employed by a large class of traditional stochastic multi arm bandit algorithms. Since the goal of the bandit algorithm is to incur low regret, to do so, it needs to figure out which arms lead to high expected rewards and then it also needs to ensure that it selects the arm with highest expected reward in most rounds. This leads to an exploration vs exploitation trade-off in the goals of the algorithm. In most cases, bandit algorithms rely on two statistics of each arm to balance the trade-off between explore and exploit: the empirical estimates on mean rewards and the corresponding variance on the estimates. The empirical means indicate which arm is likely to be the optimal, and the variances indicate how much confidence the algorithm has about its estimates. The variance of the estimate can be characterized by the number of samples the algorithm has access to for estimating the empirical means. The number of samples for each arm is exactly equal to the number of times that arm is selected by the learner in the stochastic setting. So typically, a wide class of stochastic multi arm bandit algorithms make decisions based on the empirical mean and number of selections for each arm. We call this class of algorithms as *Mean based algorithms*. Before introducing the formal definition, let us characterize the information the bandit algorithm has access to when making decisions in a round $t$. Let $\mathcal{I}^t$ denote the information the algorithm has access to while making decisions in round $t$. Using the information $\mathcal{I}^t$, the algorithm generates a probability distribution $\pi^t$ over the arms where for each arm $i$, $\pi^t(i|\mathcal{I}^t)$ is the probability that the arm $i$ is selected in the current round $t$ when the information available is $\mathcal{I}^t$.

Since in each round $t$, the algorithm chooses an arm $I^t$ and then obtains the corresponding reward $r_{I^t}^t$, the information obtained by the algorithm in round $t$ is $(I^t, r_{I^t}^t)$. Thus before making a decision in round $t$, the algorithm has access to all the information received in the rounds so far. Let us denote $H^t = \{(I^1, r_{I^1}^1), \ldots, (I^{t-1}, r_{I^{t-1}}^{t-1})\}$ as the history up till round $t$ and it is exactly the information that the bandit algorithm has access to when making the decision in this round, i.e. $\mathcal{I}^t = H^t$. Thus for the bandit algorithm, the decisions made in round $t$ can be characterized by $\pi^t(i|\mathcal{I}^t) = \pi^t(i|H^t)$.

Let $n_i^{t-1} = \sum_{\tau=1}^{t-1} \mathbb{1}\{I^\tau = i\}$ denote the number of rounds arm $i$ gets picked by the algorithm before round $t$, and let $\bar{\mu}_i^{t-1} = \frac{\sum_{\tau=1}^{t-1} r_i^\tau \mathbb{1}\{I^\tau = i\}}{n_i^{t-1}}$ be the empirical mean of the arm $i$ by round $t$. We can define *Mean based algorithms* as follows.

**Definition 3** (Mean based algorithms). *We say an algorithm is a mean based algorithm if*

*1. Its policy depends only on the empirical means $\bar{\mu}_i^{t-1}$ and number of times each arm $i$ is selected $n_i^{t-1}$ of all the arms. In other words for each arm $i$,*

$$\pi^t(i|H^t) = \pi^t(i|n_1^{t-1}, \bar{\mu}_1^{t-1}, \ldots, n_K^{t-1}, \bar{\mu}_K^{t-1})$$

*2. For each arm $i$, the probability that it is selected is monotonically increasing in its empirical mean, i.e.*

$$\pi^t(i|\ldots, n_i^{t-1}, \bar{\mu}_i^{t-1}, \ldots) \geq \pi^t(i|\ldots, n_i^{t-1}, \bar{\mu}_i'^{t-1}, \ldots)$$

*if $\bar{\mu}_i^{t-1} \geq \bar{\mu}_i'^{t-1}$*

*3. For each sub-optimal arm $i$, the probability that it is selected is monotonically decreasing on number of selections, i.e*

$$\pi^t(i|\ldots, n_i^{t-1}, \bar{\mu}_i^{t-1}, \ldots) \leq \pi^t(i|\ldots, n_i'^{t-1}, \bar{\mu}_i^{t-1}, \ldots)$$

*if $n_i^{t-1} \geq n_i'^{t-1}$ and $\bar{\mu}_i^{t-1} < \max \bar{\mu}_{j \in [K]}^{t-1}$.*

In Definition 3, condition 1. implies that the algorithm's decisions only depends on the empirical mean and the number of pulls of each arm so far. Condition 2. implies that if the empirical mean of the arm is higher, if every other statistic remains the same, then the probability that the arm gets selected only increases. Condition 3. implies that if the arm is empirically sub-optimal, then if the number of samples used to obtain that estimate increases, then the algorithm is more confident about the fact the arm is sub-optimal, then the probability that the arm gets selected can only decrease.

Many classical bandit algorithms such as UCB, $\epsilon$-greedy, and Thompson sampling fall into the framework of *mean based algorithms*. In the next section we introduce our attack methodology using

the adversary in consideration. Using the attack, we can show that all mean based algorithms are vulnerable to data corruptions attacks. In subsequent sections we prove stronger guarantees for a number of classical multi arm bandit algorithms by showing that UCB, $\epsilon$-greedy, and Thompson Sampling algorithms are completely vulnerable to our attacks as long as the mean reward of the target arm is not too small.

## 3  Observation-Free Attack

In this section we introduce a data poisoning attack that we call the Observation-Free Attack (Algorithm 2) which doesn't explicitly observe the behavior of the bandit algorithm while deciding how to corrupt rewards.

The attack is separated into three phases. In the first phase that lasts for $C_1$ rounds, the attack aims at making the algorithm receive a lot of low rewards from the optimal arm so that the empirical estimate of the optimal arm's mean reward is as low as possible and that the confidence of the algorithm over its estimate is high. To ensure that the optimal arm is picked enough times, we attack all arms which makes all arms appear equally bad to the algorithm. Explicitly, we set reward to be 0 for all arms in all the rounds in the first phase.

In the second phase that lasts for $C_2$ rounds, the attack tries to make the target arm distinguishable from the other arms. That is, it wants the algorithm to think that the empirical reward of the target arm is much better than all other arms. The corresponding way is to set the reward as 1 for that target arm and 0 for all other arms. Let $\tilde{i}$ be target arm, then the corrupted reward $\hat{r}^t$ in second phase is set as $e_{\tilde{i}} \in [0,1]^K$ where $e_{\tilde{i}}$ is the vector with 1 at the index $\tilde{i}$ and 0 everywhere else. By the end of the first two phases, the adversary has tried to ensure that empirical mean of all arms except the target arm is very low with high confidence and that the empirical mean of the target arm is much higher than the other arms.

In the third phase, the adversary does nothing and hopes that the algorithm selects the target arm for most of the rounds and no other arm can recover from the initial corruption applied to their rewards in the first two phases. So the attack only corrupts the initial $C_1 + C_2$ rounds and the corruption level is $C_1 + C_2$.

---

**Algorithm 1:** Observation-Free Attack

**Parameters :** Number of rounds $T$, Mean rewards vector $\bar{\mu}$, bandit algorithm $A$, target arm $i$

1 Compute parameters $C_1$ and $C_2$ for the given $T, \bar{\mu}, A$.
2 **for** $t = 1, \ldots, T$ **do**
3      Environment generates the reward vector $r^t$
4      **if** $t \leq C_1$ **then**
5         $\hat{r}^t \leftarrow (0, \ldots, 0)$                       /* Set reward as 0 for all arms */
6      **end**
7      **else if** $C_1 < t \leq C_1 + C_2$ **then**
8         $\hat{r}^t \leftarrow e_{\tilde{i}}$       /* Set reward as 0 for all arms but the target arm.  The reward for the target arm is 1 */
9      **end**
10     **else**
11        $\hat{r}^t \leftarrow r^t$                              /* No corruption is applied */
12     **end**
13     Bandit algorithm $A$ selects arm $I^t$ and receives reward $\hat{r}^t_{I^t}$
14 **end**

---

$C_1$ and $C_2$ are the two parameters that the adversary needs to tune based on the bandit algorithm under consideration and the rewards of the arms. For the sake of analysis, we assume that adversary has access to the mean reward for each of the arms, i.e the adversary knows $\boldsymbol{\mu} = (\mu_1, \ldots, \mu_K)$ before the start of the bandit learning algorithm. If the adversary has access to the mean rewards, then the adversary doesn't even need to access the realized rewards from any of the rounds to decide its strategy. If the adversary does not have access to the mean rewards before the start of the process, then we show in appendix 7 that while corrupting the first few rounds, the adversary can observe the

realized rewards to effectively estimate the mean rewards. Using the estimates, the adversary can set the parameters $C_1$ and $C_2$ of Algorithm 2 in an adaptive manner.

---

**Algorithm 2:** Bandit learning with data poisoning attack

---
**Parameters:** Number of rounds $T$, bandit algorithm $A$, adversary $M$
1 **for** $t = 1, \ldots, T$ **do**
2     Environment generates the reward vector $r^t$
3     **if** *Weak attack* **then**
4        Adversary $M$ replace the reward vector by $\hat{r}^t$
5     **end**
6     Bandit algorithm $A$ selects arm $I^t$
7     **if** *Strong attack* **then**
8        Adversary $M$ observe $I^t$ and replace the reward vector by $\hat{r}^t$
9     **end**
10     Bandit algorithm $A$ receives reward $\hat{r}_{I^t}^t$
11 **end**

---

## 4    Vulnerability of Mean Based Bandit Algorithms

In this section we show the main result of this paper that all mean based bandit algorithms are vulnerable. In another word, any algorithm that only makes decisions that depend only on the empirical means of the arms so far and the number of time each arm has been pulled so far are not robust.

**Theorem 1.** *For any mean based bandit algorithm that achieves sub-linear regret in the absence of data-corruptions, there always exists an instance with an adversary data corruption attack such that the algorithm will suffer linear regret $R(T) = \Omega(T)$ in expectation.*

To prove the theorem, we show there exist three instances such that the algorithm must suffer linear regret in at least one of the three instances. We apply observation free attack in the first instance. In the second instance, we only attack the first few rounds and show that algorithm either suffers from linear regret in this instance, or almost always picks the target arm at the second phase in the first instance. In the third instance, we apply no attack and show that either the algorithm suffers from linear regret in this instance, or only picks the optimal arm for a few rounds at the third phase in the first instance. Then if the algorithm guarantees sub-linear regret in the second and the third instance, then it must suffer from linear regret in the first instance.

Here we provide an intuition for why mean based algorithms are vulnerable. Mean based algorithms make decisions based on estimates on arms mean value and error from variance. However, the adversary could introduce additional bias to the estimates which is unknown to and omitted by the algorithms. Such bias could keep the estimates far from the real value for most of time through only slight corruption, hence the algorithm will always make poor decisions, which leads to big regret.

So far we have shown that the observation free attack can induce linear regret on the algorithm in some instances with $\Omega(1)$ probability if such algorithm perform well in some other instances. Actually, the observation free attack is more powerful when attacking some specific mean based algorithms. In the next section we will show that UCB, $\epsilon$-greedy, and Thompson sampling algorithms are completely vulnerable to the attack, that is, as long as the target arm has $\Omega(1)$ mean reward, the adversary with low corruption level is able to manipulate the bandit algorithm to almost always pick the target arm with high probability. Also, note that the famous EXP3 algorithm is robust in this setting as it can work even in the fully adversarial setting which includes this setting as a special case. Unlike the other classical algorithms we have just mentioned, EXP3 algorithm is not a mean-based algorithm as it doesn't use the empirical mean of rewards to make decisions.

## 5    Attack on Stochastic Bandit Algorithms

In this section we analyze the performance of the Observation-Free attack on different classical stochastic multi arm bandit algorithms including UCB, $\epsilon$-greedy, and Thompson sampling algorithms.

We show how we can tune the parameters $C_1$ and $C_2$ for each of the algorithm and present the corresponding guarantees on the vulnerability of the algorithms when subjected to our attacks.

## 5.1 Attack on UCB Algorithm

The UCB algorithm [Auer et al., 2002] is probably the most popular stochastic multi arm bandit algorithm. UCB works by maintaining upper confidence bounds on the empirical means of the arms' rewards and chooses the arm with the highest UCB value in each round. Formally, the arm selection rule of a standard UCB algorithm is the following.

$$I^t = \begin{cases} t, & \text{if } t \leq K \\ \text{argmax}_i\{\bar{\mu}_i^{t-1} + \sqrt{\frac{\log T}{n_i^{t-1}}}\}, & \text{otherwise} \end{cases} \tag{1}$$

where $\bar{\mu}_i^{t-1}$ and $n_i^{t-1}$ are the empirical mean and number of times selected so far for arm $i$ by round $t$. Ties can be broke arbitrarily. Let arm $i^*$ be the optimal arm, and arm $\tilde{i}$ be the target arm. Let $\mu = \mu_{\tilde{i}}$ denote the mean reward of the target arm for the rest of the paper.

**Theorem 2.** *When an adversary applies data corruption attack on UCB algorithm with the attack given by Algorithm 2, by choosing appropriate $C_1$ and $C_2$, with corruption level $C = O(\frac{K \log T}{\mu^2})$ where $\mu$ is the mean reward of the target arm, the UCB algorithm pulls the target arm for all but $O(\frac{K \log T}{\mu^2})$ rounds with probability at least $1 - 1/T$.*

The proof ideas for the analysis of attack on UCB algorithm and the other two algorithms mentioned later this section are similar. During the first stage where $t \leq C_1$, each arm will get selected for around $C_1/K$ rounds and the empirical mean for all arms will be 0. During the second phase where $C_1 < t \leq C_1 + C_2$, the adversary starts injecting high reward for the target arm and still keeps corrupting the other arms' rewards to 0. The target arm will have the highest mean and thus will get picked most frequently. $C_2$ is chosen to be big enough such that the empirical mean of target arm will never be lower than its true mean with high probability. At the end of the second phase, all arms other than the target arm have been corrupted heavily. During the last stage where $t > C_1 + C_2$, since the target arm has a high enough empirical mean, it gets picked the most often. By choosing $C_1$ and $C_2$ appropriately, we can ensure that even if the other arms are explore in the third phase, they get picked so infrequently that their empirical mean cannot recover by the end of the $T$ rounds to be better than that of the target arm. Thus, the target arm will be empirically optimal arm throughout the last phase and thus will be chosen the most often.

## 5.2 Attack on $\epsilon$-greedy Algorithm

In $\epsilon$-greedy Algorithm, with some probability $\epsilon$, the algorithm decides to randomly select an arm to *explore*. Otherwise, the algorithm picks the am which is empirically best so far. Formally, the arm-selection rule of $\epsilon$-greedy algorithm with an explore rate $\epsilon$ is:

$$I^t = \begin{cases} \text{draw uniform[K]}, & \text{w.p.} \epsilon \\ \text{argmax}_i\{\bar{\mu}_i^{t-1}\}, & \text{otherwise} \end{cases} \tag{2}$$

**Theorem 3.** *When an adversary applies data corruption attack on $\epsilon$-greedy algorithm with the attack given by algorithm 2, by choosing appropriate $C_1$ and $C_2$, with corruption level $C = \tilde{O}(T\epsilon/\mu + K)$ where $\tilde{O}$ hides $\log T$ terms and $\mu$ is the mean reward of the target arm, the $\epsilon$-greedy algorithm pulls the target arm for all but $\tilde{O}(T\epsilon/\mu) + K)$ rounds with probability at least $1 - \frac{2K+2}{T}$.*

For $\epsilon$-greedy algorithm, in the absence of corruption, appropriate choice of $\epsilon$ is important to ensure sub-linear regret. The $T\epsilon$ term in unavoidable in the regret of epsilon greedy thus to ensure sub-linear regret in the absence of corruptions, the $\epsilon$ chosen by the learner has to be such that $T\epsilon$ is sub-linear. This also implies that our corruption level is also sub-linear. A typical choice is $\epsilon = O(T^{2/3})$, then the corresponding level for the attack is $\tilde{O}(T^{2/3}/\mu + K)$, and the target arm will be selected for all but $\tilde{O}(T^{2/3}/\mu + K)$ rounds with probability at least $1 - \frac{2K+2}{T}$.

## 5.3 Attack on Thompson Sampling Algorithms

Here we analyze the Thompson sampling algorithm on Bernoulli Bandits with Beta Distribution as posterior distribution [Agrawal and Goyal, 2012]. In this setting, the reward from picking an arm $i$ in any round is a Bernoulli random variable with mean $\mu_i$. Let $\bar{\mu}_i$ be the empirical mean reward of arm $i$ and $n_i$ be the number of rounds when arm $i$ is picked. At round $t$, for every arm $i$, the algorithm samples $\theta_i^t$ from the posterior distribution $B(\bar{\mu}_i \cdot n_i + 1, (1 - \bar{\mu}_i) \cdot n_i + 1)$ associated with the arm. Here $B(\cdot, \cdot)$ is a beta distribution. Then the algorithm chooses the arm with the highest sampled value, that is, $I^t = \text{argmax}_i \theta_i^t$.

**Theorem 4.** *When an adversary applies data corruption attack on the Thompson sampling algorithm with the attack given by algorithm 2, by choosing appropriate $C_1$ and $C_2$, with corruption level $C = O(\frac{2^K \log T}{\mu^2})$ where $\mu$ is the mean reward of the target arm, the Thompson sampling algorithm will pull the target arm for all but $O(\frac{2^K \log T}{\mu^2})$ rounds with probability at least $1 - \frac{2K+1}{T}$.*

The theorems in this section conclude that as long as $\frac{1}{\mu^2}$ is sub linear in $T$ where $\mu$ is the mean reward for the target arm, then an adversary using the observation free attack that ensure can the algorithms picks a target arm of their choice for all but $o(T)$ rounds with high probability. In the following section, we experimentally evaluate the performance of the difference algorithms when subjected to the observation free attack.

# 6 Experiments

In this section, to intuitively illustrate the behavior of algorithms under corruption by our adversary algorithm, we run simulations attacking UCB, $\epsilon$-greedy and Thompson Sampling algorithm. Each algorithm is tested under the same artificial instance with 2 arms, with means $\mu_1 = 0.9$ and $\mu_2 = 0.8$. The arm 1 is the optimal arm and we set arm 2 as the target arm for the adversary. We set $T = 50000$ and the corresponding parameters $(C_1, C_2)$ for each of the algorithm is listed in Table 1.

| Algorithm | $C_1$ | $C_2$ |
|---|---|---|
| UCB | 34 | 66 |
| $\epsilon$-greedy | 150 | 150 |
| Thompson Sampling | 34 | 66 |

Table 1: Corruption level parameters for different algorithms

In Figure 1, we plot some key statistics about the arms as a function of the iterations that can help us understand the behaviour of the algorithms under the attacks. In Figure 2, we plot the number of times the optimal arm is pulled is chosen till round $t$, i.e. $n_{i*}^t$ with the iteration $t$ on the x axis in both the settings. We consider the case when there is no attack and how the number changes when we do attack the algorithm. In both Figure 1 and Figure 2, the top row zooms in on the iterations in phase 1 and 2, i.e. the corrupted rounds whereas the bottom row shows the behaviour till the horizon $T$.

**UCB Algorithm**

In UCB algorithm, the main statistic used by the algorithm is the UCB on the arms' mean reward. In each round, the arm with the highest UCB value is picked. In sub-figures $(a1)$ and $(a2)$ in Figure 1, we plot the UCB values for both the target arm and optimal arm. We can see sub-figures $(a1)$ that in the first phase, i.e. $t \leq C_1$ the UCB value for both the arms decreases to a value close to 0. Then in the next phase as we start injecting high rewards for the target arm, the UCB value for the target arm grows but it remains close to 0 for the optimal arm. In the third phase, after the corruption rounds, in sub-figures $(a2)$ we can see that till the end of the horizon, UCB value of the target arms remains greater than that of the optimal arm. Even the mean of the target arm decreases towards in the direction of the real mean, it never fall below the UCB of the optimal arm. In sub-figures $(a1)$ and $(a2)$ of Figure 2, we plot the the number of cumulative times the optimal arm gets pulled by the round $t$. In sub-figure $(a1)$ of Figure 2, we can see that in the second phase, as we start injecting higher rewards in the target arm, the algorithm completely stops choosing the optimal arm. After the second phase also, we can see in sub-figure $(a2)$ of Figure 2 that the optimal arm never almost never gets pulled. In the absence of corruptions, UCB algorithm performs very well and the optimal arm is pulled almost always.

### $\epsilon$-greedy Algorithm

In $\epsilon$-greedy Algorithm, the key value to an arm's performance is its empirical mean. When there are two arms, the arm with higher empirical mean will be picked with probability $1 - \epsilon/2$. In sub-figures $(b1)$ and $(b2)$ of Figure 1, we plot the empirical mean for both the target arm and optimal arm. Similar to UCB we see than in Phase 1, the empirical means concentrate around 0, then the empirical mean for target arm increases in phase 2, and then the target arm remains the empirically optical arm till the end of horizon. Similar behaviour is seen in the number of times the optimal arm gets pulled. We see in sub-figures $(b1)$ and $(b2)$ of Figure 2 that under the attack, after Phase 1, the optimal arm gets picked very infrequently (only in explore rounds) whereas in the absence of corruptions, the optimal arm is picked almost always.

### Thompson sampling Algorithm

In Thompson Sampling algorithm, the algorithm maintains a Beta distribution for each arm. Based on the Beta distribution for the two arms, in sub-figures $(c1)$ and $(c2)$ of Figure 1, we plot the approximate probability that a sample from the empirical Beta distribution associated with the optimal arm is greater than a sample from the empirical Beta distribution of the target arm. Again, similar to UCB, we can see that in sub-figure $(c1)$ of Figure 1 that after Phase 1, the probability that the optimal arm is chosen drops close to zero. In sub-figure $(c2)$ of Figure 1, we observe that the optimal arm can never recover from the corruption and the probability that it gets selected remains close to 0. This is reflected in sub-figures $(c1)$ and $(c2)$ of Figure 2 where we can see that under attack, after phase 1, the optimal arm never gets picked whereas in the absence of corruptions, the optimal arm is picked almost always.

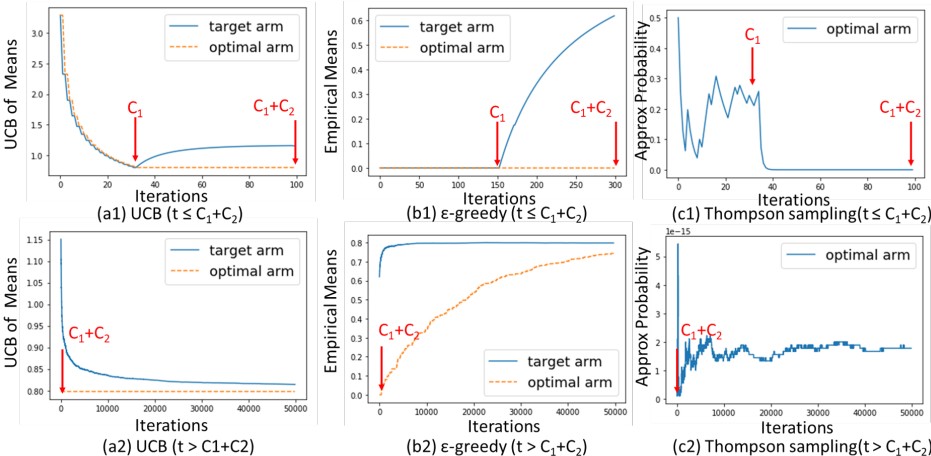

Figure 1: Empirical behaviors of arms in different algorithms. (a), (b) is for UCB algorithm; (c), (d) is for $\epsilon$-greedy algorithm; (e), (f) is for Thompson sampling algorithm. (a), (c), (e) focus on the time when the rewards are being corrupted. (b), (d), (f) focus on the time when the attack stops.

To intuitively show how different algorithms behave with and without the existence of adversary, we plot the counts of the number of rounds the optimal arm get picked versus time in figure 2.

## 7 Attack agnostic to mean rewards of arms

We assumed in Section 3 that the adversary has access to mean rewards of each arm which is required to set the parameters of Algorithm 2. We can introduce a slight modification on the original attack such that the new attack can be agnostic to the mean rewards while maintaining similar performance.

The modified observation free attack works as follows. The attack is still separated into three phases and applies corruption in the same way as before. At the beginning $C_1$ is set to be infinite so that the attack can estimate the mean reward $\mu$ of the target arm, and once an accurate estimate is formed, the attack can set $C_1$ and $C_2$ based on the estimate. The question is how to decide the time $\tau$ when the estimating ends. Here is some intuition how we set $\tau$. Let $n^t$ denote the number of rounds the target arm gets selected by round $t$. The adversary can have a lower confidence bound on the mean

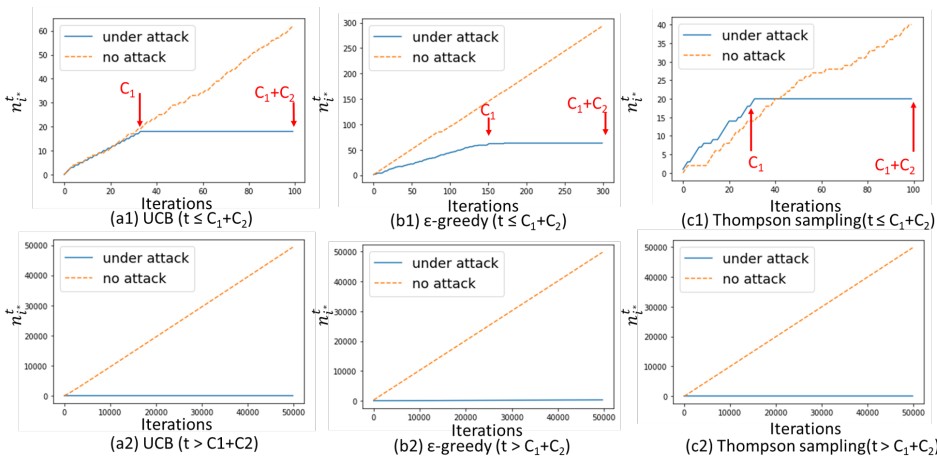

Figure 2: The number of rounds the optimal arm get. (a1), (a2) is for UCB algorithm, (b1), (b2) is for $\epsilon$-greedy algorithm, and (c1), (c2) is for Thompson sampling algorithm.

reward of the target arm as $\mu_{\text{LCB}} = \bar{\mu} - \sqrt{\frac{\log T}{n_t}}$. By Hoeffding inequality, with probability at least $1 - 2/T^2$, we have $\bar{\mu} \in [\mu - \sqrt{\frac{\log T}{n_t}}, \mu + \sqrt{\frac{\log T}{n_t}}]$, which implies $\mu_{\text{LCB}} \in [\mu - 2\sqrt{\frac{\log T}{n_t}}, \mu]$. Note that $\sqrt{\frac{\log T}{n_t}}$ diminishes from positive from infinite to 0 as $n_t$ grows, so there exists a turning $n_t^*$ such that $\sqrt{\frac{\log T}{n_t^*}} < \mu/4$ and $\sqrt{\frac{\log T}{n_t^*-1}} > \mu/4$. Based on this fact, the attack can stop estimating when $\mu_{\text{LCB}} \geq 2\sqrt{\frac{\log T}{n_t}}$ becomes true. At this time, with probability at least $1 - 2/T$, $\mu_{\text{LCB} \in [\mu/2, \mu]}$, in another word, $\mu_{\text{LCB}=O(\mu)}$. Then the attack can set $C_1$ and $C_2$ by setting the mean reward for the target arm as $\mu_{\text{LCB}}$. If the time $\tau$ to set $C_1$ is already greater than $C_1$, then let $C_1 = \tau$ and determine new $C_2$ based on the new $C_1$ and $\mu_{\text{LCB}}$ correspondingly.

**Lemma 5.** *When attacking UCB algorithm with the new attack, with corruption level $C = O(\frac{K \log T}{\mu^2})$, the UCB algorithm will pull the target arm for all but $O(K \log T/\mu^2)$ rounds with probability at least $1 - 3/T$. When attacking $\epsilon$-greedy algorithm, with corruption level $C = \tilde{O}(T\epsilon + K)$, the $\epsilon$-greedy algorithm will pull the target arm for all but $\tilde{O}(T\epsilon + K)$ rounds with probability at least $1 - 2K + 4/T$. When attacking UCB algorithm with the new attack, with corruption level $C = O(\frac{2^K \log T}{\mu^2})$, the Thompson sampling algorithm will pull the target arm for all but $O(\frac{2^K \log T}{\mu^2})$ rounds with probability at least $1 - 3/T$.*

As we show earlier, with probability at least $1 - 2/T$, the true mean reward of the target arm satisfies $\mu \in [\bar{\mu} - \sqrt{\frac{\log T}{n_t}}, \bar{\mu} + \sqrt{\frac{\log T}{n_t}}]$. If this is true, then when the adversary determines $C_1$ and $C_2$, $\sqrt{\frac{\log T}{n_t}} \geq \mu/2$, which is equivalent to $n_t \leq \frac{4 \log T}{\mu^2}$. Note that in all the three algorithms mentioned above, $n_t$ is at least $t/K - \sqrt{t \log T}$ with probability at least $1 - 1/T$. So the time when the adversary determine $C_1$ and $C_2$ is at most $\frac{16K \log T}{\mu^2} + K^2 \log T$, and $\mu_{LCB} \geq \mu/2$. Since $\mu_{LCB} \leq \mu$, we have $\mu_{LCB} = \Omega(\mu)$. If this $C_1$ is larger than the current time $\tau$, the algorithm will stay in phase 1 until $C_1$ and behave exactly the same as the old attack with $\mu$ replaced by $\mu_{\text{LCB}}$, which results in corruption level with the same order. If this $C_1$ is less than the current time $\tau$, then the algorithm will set $C_1 = \tau$ instead. In this case $\tau$ is of the same order as $C_1$ of the old attack since $\tau = O(C_1)$ for $C_1$'s in attacking all algorithms, the corruption level the new attack needs is still of the same order of the old one, and the probability that the new attack would fail is $2/T$ greater than the old one because of the chance that the estimation of $\mu$ is inaccurate.

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
