# A Appendix

We now prove the main result about the vulnerability of mean based algorithms (Theorem 1). That is, for any mean based bandit algorithm that achieves sub-linear regret in the absence of data-corruptions, there always exists an instance where an adversarial data corruption attack with $o(T)$ corruption level can make the algorithm suffer linear regret $R(T) = \Omega(T)$ in expectation.

## A.1 Proof for Theorem 1

*Proof.* Denote the two arms in instances with two arms as $a_1$ and $a_2$. Given an instance where the means of both arms are 0. For any constant $C_1$, there is always at least one arm such that it gets at least $C_1/2$ picks with probability at least $1/2$, denote such arm as $a_1$. We consider an instance (1) where $a_1$ is the optimal arm:

$$\mu_{a_2}^{(1)} = \mu,$$
$$\mu_{a_1}^{(1)} > \mu_{a_2}^{(1)}.$$

We will perform the observation free attack on instance (1). In the first phase of attack, the rewards are always 0 for any arm. By the end of the first case, for instance (1), from the way we set $a_1$, we have with probability at least $1/2$, the following will happen:

$$n_{a_1}^{(1)} \geq \frac{C_1}{2}, \hat{\mu}_{a_1}^{(1)} = 0,$$

$$n_{a_2}^{(1)} \leq \frac{C_1}{2}, \hat{\mu}_{a_2}^{(1)} = 0.$$

Let $G_1$ be the event that the above is true in instance (1), we know that $\Pr\{G_1\} \geq 1/2$.

Next, consider another instance (2) where the mean reward of $a_2$ is 1, and the mean reward of other arm is 0:

$$\mu_{a_1}^{(2)} = 0,$$
$$\mu_{a_2}^{(2)} = 1.$$

For instance (2), we corrupt the first $C_1$ rounds and set the rewards to be 0 for all arms, then stop corruption. Let $N_1^{(2)}$ be the number of rounds when the algorithm pick arm 1 after the corruption ends. Let $f_1$ be the value such that $\Pr\{N_1^{(2)} \geq f_1\} = 1/2$. The expected regret of the algorithm is at least $R^{(2)}(T) \geq 1/2 f_1$. So $f_1 \leq 2R^{(2)}(T)$, which has to be sublinear or otherwise the algorithm has linear expected regret in instance (2).

Next we focus on the second phase of attack in instance (1). Let $C_2 = f_1 + \alpha C_1$ where $\alpha$ is a parameter to be specified later. Up the end of this phase, what happened in (1) is the same as that in (2). So with probability $1/2$, $a_1$ is picked for less than $f_1$ rounds in this phase. Denote such Event as $G_2$, then $\Pr\{G_2\} = 1/2$. If both $G_1$ and $G_2$ are true, by the end of the second phase of attack, the following is true :

$$n_{a_1}^{(1)} \geq \frac{C_1}{2}, \hat{\mu}_{a_1}^{(1)} = 0,$$

$$n_{a_2}^{(1)} \geq \alpha C_1, \hat{\mu}_{a_2}^{(1)} \geq \frac{2\alpha}{2\alpha + 1}.$$

Next we focus on the last phase of attack in instance (1) where the corruption is ended. For any value of $n$, if $a_2$ get picked for $n$ times in this phase, then by Hoeffding inequality inequality, with probability at least $1 - 1/T$, the reward from these $n$ rounds is at least $\mu n - \sqrt{\log(T)n}$ for any $n \leq T$. Set $\alpha = \frac{\frac{\log(T)}{2\mu C_1} + \frac{\mu}{4}}{1 - \mu/2}$, the corresponding empirical mean of $a_2$ satisfies

$$\bar{\mu} = \frac{C_1 \cdot \alpha + n\mu - \sqrt{n\log(T)}}{C_1(\alpha + 1/2) + n} \geq \mu/2.$$

That is, in the last phase, with probability at least $1 - 1/T$, the empirical mean of $a_2$ is always greater than $\mu/2$. Let $G_3$ denote the event where the above happens, so $\Pr\{G_3\} \geq 1 - 1/T$.

Before proceeding, we introduce an instance (3) where the reward of arm $a_1$ is always $\mu/4$ and the reward of $a_2$ is always $\mu/2$. Let $n_1^t$ and $n_2^t$ be the number of rounds $a_1$ and $a_2$ get selected by round $t$. Define random variables $\{Y_1, \ldots, Y_{T/2}\}$ where $Y_n$ is $n_1^t$ if exists a $t$ such that $n_2^t = n$, and $T - n$ if such $t$ doesn't exists. It is clear that $\Pr\{Y_n < 0\} = 0$, $\Pr\{Y_n < T\} = 1$, and $\Pr\{Y_n < x\} \leq \Pr\{Y_n < x\} + 1$. So we could always find an integer $k$ such that $\Pr\{Y_{T/2} < k\} = 1/2$, and such $k$ must be sublinear in $T$ or otherwise the regret in instance (3) will be linear. $Y_n$ also satisfies $Y_n \in [Y_{n-1}, Y_{n-1} + 1, \ldots, T - n]$, and $\Pr\{Y_n = Y_{n-1} + i | Y_{n-1} = y_{n-1}\} \geq \Pr\{Y_n = Y_{n-1} + j | Y_{n-1} = y_{n-1}\}$ for all $0 \leq i \leq j$ and $y_{n-1}$. The purpose of introducing instance (3) is to show that if the algorithm have sublinear regret in this instance, then with probability $1/2$, it won't pick $a_1$ for more than $k$ times. Then in stance (1), by choosing big enough $C_1$ and $C_2$, with probability at least $1/2$, it won't pick $a_1$ for more than $k$ times, so the algorithm will have linear regret in instance (1).

Now back to instance (1) and set $C_1 = (8/\mu - 2)k$, so at the beginning of the last phase, $a_1$ has already been picked for at least $(4/\mu - 1)k$ rounds. Then the empirical mean of $a_1$ will not exceed $\mu/4$ before it get at least $k$ picks from this phase. Then by the definition of mean based algorithm, we know that before $a_1$ get its $k^{th}$ pick, the probability $a_1$ get picked in instance (1) is always less than that in instance (3) for the same number of rounds $a_2$ get picked. Let $n_1^t$ and $n_2^t$ as the number of rounds arm $a_1$ and $a_2$ get picked in the last phase by round $t$. Define random variables $\{Z_1, \ldots, Z_{T/2}\}$ in the same way as $Y_n$ where $Z^n = n_1^t$ if exists $t$ such that $n_2^t = n$ and $Z^n = T - n$ if such $t$ doesn't exists. $Z_n$ also satisfies $Z_n \in [Z_{n-1}, Z_{n-1} + 1, \ldots, T - n]$, and $\Pr\{Z_n = Z_{n-1} + i | Z_{n-1} = z_{n-1}\} \geq \Pr\{Z_n = Z_{n-1} + j | Z_{n-1} = z_{n-1}\}$ for all $0 \leq i \leq j$ and $z_{n-1}$. The relation between $Z_n$ and $Y_n$ satisfies: $\Pr\{Z_n = x | Z_{n-1} = x\} \geq \Pr\{Y_n = x | Y_{n-1} = x\}$ for all $x \leq k$ and $\Pr\{Z_n = x + i | Z_{n-1} = x\} \leq \Pr\{Y_n = x + i | Y_{n-1} = x\}$ for all $i > 0$ and $x + i \leq k$. Intuitively, $Z_n$ "grows" slower than $Y_n$ before it exceeds $k$, so $Y_n$ is more likely to reach $k$ than $Z_n$. Next are we going to strictly prove that $\Pr\{Y_{T/2} \leq k\} \leq \Pr\{Z_{T/2} \leq k\}$.

Note that $\Pr\{Y_n \leq k\}$ depends on $\Pr\{Y_m | Y_{m-1}\}$ for all $m \leq n$. The idea of the proof is to show that by substituting each $\Pr\{Y_m | Y_{m-1}\}$ by $\Pr\{Z_m | Z_{m-1}\}$, the probability of $\Pr\{Y_n \leq k\}$ will increase. We introduce another series of random variables $\{F_1^1, \ldots, F_{T/2}^1\}$ where $\{F_n^1\}$ is almost the same as $\{Y_n\}$ except that $\Pr\{F_m^1 | F_{m-1}^1\} = \Pr\{Z_m | Z_{m-1}\}$ for a specific $m$. We want to show that $\Pr\{Y_{T/2} \leq k\} \leq \Pr\{F_{T/2}^1 \leq k\}$. After that, we can construct $\{F_n^2\}$ which is almost the same as $\{F_n^1\}$ except for $\Pr\{F_{m'}^2 | F_{m'-1}^1\} = \Pr\{Z_{m'} | Z_{m'-1}\}$ where $m' \neq m$. For the same reason we will have $\Pr\{F_{T/2}^1 \leq k\} \leq \Pr\{F_{T/2}^2 \leq k\}$. Repeat this process until $\{F_n^{T/2}\}$ which is the same as $\{Z_n\}$, then we have $\Pr\{Y_{T/2} \leq k\} \leq \Pr\{F_{T/2}^1 \leq k\} \leq \Pr\{F_{T/2}^2 \leq k\} \leq \ldots \leq \Pr\{F_{T/2}^{T/2} \leq k\} = \Pr\{Z_{T/2} \leq k\}$. Next we will prove that $\Pr\{Y_{T/2} \leq k\} \leq \Pr\{F_{T/2}^1 \leq k\}$.

First, we can write $\Pr\{Y_{T/2} \leq k\}$ as

$$\Pr\{Y_{T/2} \leq k\}$$

$$= \sum_{x=0}^{k} \Pr\{Y_{T/2} \leq k | Y_{m-1} = x\} \cdot \Pr\{Y_{m-1} = x\}$$

$$= \sum_{x=0}^{k} \Pr\{Y_{m-1} = x\} \cdot \sum_{y=x}^{k} \Pr\{Y_n \leq k | Y_m = y, Y_{m-1} = x\} \cdot \Pr\{Y_m = y | Y_{m-1} = x\}$$

$$= \sum_{x=0}^{k} \Pr\{F_{m-1}^1 = x\} \cdot \sum_{y=x}^{k} \Pr\{F_n^1 \leq k | F_m^1 = y\} \cdot \Pr\{Y_m = y | Y_{m-1} = x\}$$

The difference between $\Pr\{Y_{T/2} \leq k\}$ and $\Pr\{F^1_{T/2} \leq k\}$ can be written as

$$\Pr\{Y_{T/2} \leq k\} - \Pr\{F^1_{T/2} \leq k\}$$

$$= \sum_{x=0}^{k} \Pr\{F^1_{m-1} = x\} \cdot \sum_{y=x}^{k} \Pr\{F^1_n \leq k | F^1_m = y\} \cdot (\Pr\{Y_m = y | Y_{m-1} = x\} - \Pr\{F^1_m = y | F^1_{m-1} = x\})$$

$$\sum_{y=x}^{k} \Pr\{F^1_n \leq k | F^1_m = y\} \cdot (\Pr\{Y_m = y | Y_{m-1} = x\} - \Pr\{F^1_m = y | F^1_{m-1} = x\})$$

$$= \Pr\{Y_n \leq k | Y_m = y\} \cdot (\Pr\{Y_m = x | Y_{m-1} = x\} - \Pr\{F^1_m = x | F^1_{m-1} = x\})$$

$$+ \sum_{y=x+1}^{k} \Pr\{F^1_n \leq k | F^1_m = y\} \cdot (\Pr\{Y_m = y | Y_{m-1} = x\} - \Pr\{F^1_m = y | F^1_{m-1} = x\})$$

$$= \Pr\{Y_n \leq k | Y_m = y\} \cdot \sum_{z=x+1}^{T-m} (\Pr\{F^1_m = z | F^1_{m-1} = x\} - \Pr\{Y_m = z | Y_{m-1} = x\})$$

$$+ \sum_{y=x+1}^{k} \Pr\{F^1_n \leq k | F^1_m = y\} \cdot (\Pr\{Y_m = y | Y_{m-1} = x\} - \Pr\{F^1_m = y | F^1_{m-1} = x\})$$

$$\leq \Pr\{Y_n \leq k | Y_m = y\} \cdot \sum_{z=x+1}^{y} (\Pr\{F^1_m = z | F^1_{m-1} = x\} - \Pr\{Y_m = z | Y_{m-1} = x\})$$

$$+ \sum_{y=x+1}^{k} \Pr\{F^1_n \leq k | F^1_m = y\} \cdot (\Pr\{Y_m = y | Y_{m-1} = x\} - \Pr\{F^1_m = y | F^1_{m-1} = x\})$$

$$= \sum_{y=x+1}^{k} (\Pr\{F^1_n \leq k | F^1_m = x\} - \Pr\{F^1_n \leq k | F^1_m = y\})(\Pr\{F^1_m = y | F^1_{m-1} = x\} - \Pr\{Y_m = y | Y_{m-1} = x\})$$

We can directly have $\Pr\{F^1_m = y | F^1_{m-1} = x\} - \Pr\{F^1_m = y | F^1_{m-1} = x\} \leq 0$, for the other term, we have:

$$\Pr\{F^1_n \leq k | F^1_m = x\}$$
$$= \Pr\{F^1_n \leq k | F^1_{m+1} \leq k, F^1_m = x\} \cdot \Pr\{F^1_{m+1} \leq k | F^1_m = x\}$$
$$= \Pr\{F^1_n \leq k | F^1_{m+1} \leq k\} \cdot \Pr\{F^1_{m+1} \leq k | F^1_m = x\}$$
$$\geq \Pr\{F^1_n \leq k | F^1_{m+1} \leq k\} \cdot \Pr\{F^1_{m+1} \leq k | F^1_m = y\}$$
$$= \Pr\{F^1_n \leq k | F^1_m = y\}$$

So eventually we have

$$\Pr\{Y_{T/2} \leq k\} - \Pr\{F^1_{T/2} \leq k\}$$

$$= \sum_{x=0}^{k} \Pr\{F^1_{m-1} = x\} \cdot \sum_{y=x}^{k} \Pr\{F^1_n \leq k | F^1_m = y\} \cdot (\Pr\{Y_m = y | Y_{m-1} = x\} - \Pr\{F^1_m = y | F^1_{m-1} = x\})$$

$$\leq \sum_{x=0}^{k} \Pr\{F^1_{m-1} = x\} \cdot \sum_{y=x+1}^{k} (\Pr\{F^1_n \leq k | F^1_m = x\}$$

$$- \Pr\{F^1_n \leq k | F^1_m = y\})(\Pr\{F^1_m = y | F^1_{m-1} = x\} - \Pr\{Y_m = y | Y_{m-1} = x\})$$
$$\leq 0$$

As discussed before, by the same process we have $\Pr\{F^1_{T/2} \leq k\} - \Pr\{F^2_{T/2} \leq k\} \leq 0$ and so on. So $\Pr\{Y_{T/2} \leq k\} \leq \Pr\{F^1_{T/2} \leq k\} \leq \Pr\{F^2_{T/2} \leq k\} \leq \ldots \leq \Pr\{F^{T/2}_{T/2} \leq k\} = \Pr\{Z_{T/2} \leq k\}$. Next we will prove that $\Pr\{Y_{T/2} \leq k\} \leq \Pr\{F^1_{T/2} \leq k\}$. That is, with probability at least $1/2$, in instance (1), $a_2$ will be picked for more than $T/2$ rounds and by that time $a_1$ is picked for less than $k$ rounds.

Suppose the algorithm guarantee sublinear regret in instances (2) and (3). Let $\mu = 1/2$ and the mean reward of the optimal arm as 1, set $C_1 = 14k$ and $C_2 = f_1 + \frac{3}{4}\log(T) + \frac{7}{3}k$, the expected regret for the algorithm in instance (1) is at least $T/16$. $\qquad\square$

### A.2 Proof for Theorem 2

*Proof.* Let 1 as the index of the target arm. Under the adversarial attack, in the first phase of attack when $t \leq C_1$, the empirical mean of any arm will always be 0, so the empirical upper confidence for each arm $j$ satisfies

$$\mathrm{UCB}^t_j = \hat{\mu}^t_j + \sqrt{\frac{\log T}{n^t_j}} = \sqrt{\frac{\log T}{n^t_j}}.$$

It is clear that $\mathrm{argmax}_j \mathrm{UCB}^t_j = \mathrm{argmin}_j n^t_j$. So an arm could get its $n+1^{th}$ pick only after all other arms get selected at least $n$ times. That is, arms will be selected in turn. Hence, when $t = C_1 + 1$, all arms will be selected for $C_1/K$ times.

In the second phase of attack When $C_1 < t \leq C_1 + C_2$, the empirical mean of the target arm is increasing whenever it get selected while that of the others remain 0. If we choose $C_1 \geq \frac{4\log T}{K}$, then the upper confidence bound of the target arm 1 when it gets $n$ picks at this period satisfies:

$$\mathrm{UCB}^t_1 = \hat{\mu}^t_1 + \sqrt{\frac{\log T}{n^t_1}}$$

$$= \frac{n}{n + 4\log T/K^2} + \sqrt{\frac{\log T}{C_1/K + n}}$$

$$\geq \sqrt{\frac{\log T}{C_1/K}} = \mathrm{UCB}^t_{i \neq 1}$$

So the target arm will get all the $C_2$ picks at this period. We choose $C_1 = \max\{\frac{K\log(T)}{\mu^2_1}, \frac{4\log T}{K}\}$, so that the upper bound of other arms at the end of the second phase will be no greater than $\mu_1$. Considering the fact that $K \geq 2$ and $\mu_1 \leq 1$, we have $C_1 = \frac{K\log(T)}{\mu^2_1}$. Then we choose $C_2 = \frac{\mu_1}{1-\mu_1}C_1$ so that at the end of the second phase, the empirical mean of the second arm is its true mean $\mu_1$

In the last phase of attack when $t > C_1 + C_2$, we will show that the target arm will be picked for all rounds with a high probability. When the target arm get $n$ picks in this phase, by Hoeffding inequality, with probability at least $1 - 1/T$, the total reward generated from these $n$ rounds is greater than $\mu_1 n - \sqrt{n\log T}$ for any value of $n < T$. Denote the number of rounds the target arm get picked before $t = C_1 + C_2$ as $m$, then the upper bound of the target arm satisfies

$$\mathrm{UCB}^t_j = \hat{\mu}^t_1 + \sqrt{\frac{\log T}{n^t_1}} \geq \mu_1 - \frac{\sqrt{n\log T}}{n+m} + \sqrt{\frac{\log T}{n+m}} > \mu_1.$$

Therefore the target arm's upper confidence bound is always the highest no matter how many times it get picks in the last phase, which means it will always get picked with probability at least $1 - 1/T$.

In conclusion, to defeat UCB algorithm, the observation free attack corrupt the first $\max\{\frac{K\log(T)}{\mu^2_1}, \frac{4\log T}{K}\}/(1-\mu_1)$ rounds, and the number of rounds arm other than the target get selected is less than $\frac{(K-1)\log(T)}{\mu^2_1}$ with probability at least $1 - \frac{1}{T}$. $\qquad\square$

## A.3 Proof for Theorem 3

*Proof.* We refer to the rounds where the algorithm randomly pick an arm from all arms as "explore" rounds. Under the corruption from adversary algorithm, in the first phase of attack when $t < C_1$, all arms have the same probability to get picked because their empirical means are all $0$. So each arm will get picked no less than

$$n_1 = C_1/K - \sqrt{C_1 \log T}$$

rounds and no more than

$$n_2 = C_1/K + \sqrt{C_1 \log T}$$

with probability at least $1 - \frac{K}{T}$ given by Hoeffding inequality. Next we will discuss the case where the above is true.

In the second phase of attack when $C_1 < t \le C_1 + C_2$, once the target arm get one pick, its empirical mean will be the highest, and it will be selected with probability at least $1 - \epsilon$. With probability at least $1 - 1/T$, the target arm will get its first pick after $K \log(T)$ rounds. After that, with probability at least $1 - 1/T$, the target arm will get picked for at least

$$n(C_2) = (C_2 - K \log(T))(1 - \epsilon) - \sqrt{C_2 \log(T)}$$

times. Denote $\mu$ as the empirical mean of the target arm, to simplify the analysis, we choose $C_2$ big enough such that the target arm can get picked at least $n_3 = \max\{\frac{\log T}{\mu^2}, n_2 \frac{\mu}{1-\mu}\}$ times during this period. The reason we choose this $n_3$ is to make sure that the empirical mean of target arm is high enough when $t > C_1 + C_2$, which will be shown later. To make sure $n(C_2) \ge n_3$, we can choose

$$C_2 = K \log T + \frac{2n_3}{1 - \epsilon}$$

.

In the last phase of attack when $t > C_1 + C_2$, we want to find a lower bound on empirical mean of the target arm. Note that $n_3 \ge \frac{4 \log T}{\mu^2}$, so that empirical mean of the target arm at the beginning of this phase $t = C_1 + C_2 + 1$ is greater than $\mu$. Denote the number of rounds the target arm get picked after $t = C_1 + C_2$ as $m$, the empirical mean of the target arm satisfies:

$$\hat{\mu} \ge \frac{\mu n_3 + \mu m - \sqrt{m \log T}}{n_3 + m}$$

$$= \mu - \frac{\sqrt{m \log T}}{n_3 + m}$$

$$\ge \mu - 0.5 \sqrt{\frac{\log T}{n_3}}$$

$$= 0.5\mu$$

Therefore, before an arm other than the target arm has its empirical mean greater than $0.5\mu$, the probability it get picked is $\epsilon/K$. We want $C_1$ to be big enough such that the empirical means of other arm are always less than $\mu/2$ in the last phase. From Hoeffding inequality, with probability at least $1 - 1/T$, an arm will get picked from explore rounds for at most $T \log T \epsilon/K$ rounds. If the arm never get picked from the exploit rounds, its empirical mean satisfies:

$$\hat{\mu}_i \le \frac{T \log T \epsilon/K}{T \log T \epsilon/K + n_1}.$$

Set

$$C_1 = T \log T \epsilon(4/\mu - 2),$$

such that

$$n_1 = (T \log T \epsilon/K)(2/\mu - 1),$$

then we have $\hat{\mu}_i \le \mu/2$. So with this $C_1$, with probability at least $1 - K/T$, the empirical mean of other arms never exceed that of the target arm hence get not picks from the explore rounds. Based on such $C_1$, the corresponding $C_2$ is

$$C_2 = K \log T + \frac{2}{1-\epsilon}(\max\{\frac{\log T}{\mu^2}, \frac{\mu}{1-\mu}(C_1/K + \sqrt{C_1 \log T})\})$$

With such $C_1$ and $C_2$, the $\epsilon$-greedy algorithm will pick arms other than the target arm by at most $C_1 + T\epsilon + \sqrt{C_2 \log T}$ times with probability at least $1 - (2K+2)/T$.

$\square$

### A.4 Proof for Theorem 4

*Proof.* Let 1 be the index of the target arm. When $t < C_1$, we want to show that all arms will get picked for around $C_1/K$ rounds. Let's start with the case where $K = 2$. Denote $\Delta^t$ as the difference of number of rounds the other get picked, and $\Delta^{t+1} - \Delta^t$ as $\delta^t$. The probability that the arm which get more picked before get picked this round is no greater than $1/2$. That is, if $\Delta^t \geq 0$, $\Pr\{\delta^t = 1\} \leq 1/2$ and $\Pr\{\delta^t = -1\} \geq 1/2$; if $\Delta^t \leq 0$, $\Pr\{\delta^t = 1\} \geq 1/2$ and $\Pr\{\delta^t = -1\} \leq 1/2$. Since $\Delta^{t=C_1+1} = \sum_{t=1}^{C_1} \delta^t$, with probability at least $1 - 1/T$, $\Delta^{t=C_1+1} \leq \sqrt{C_1 \log T}$. In the case where $K > 2$, we can define $\Delta_{i,j}^t$ and $\delta_{i,j}^t$ as the $\Delta^t$ and $\delta_{i,j}^t$ arm $i$ and $j$, and by similar argument we have with probability at least $1 - 1/T$, $\Delta_{i,j}^{t=C_1+1} \leq \sqrt{C_1 \log T}$. This means at round $t = C_1 + 1$, with probability at least $1 - K/T$, the number of rounds any arm get picked is no less than

$$n_1 = \frac{C_1 - (K-1)\sqrt{C_1 \log T}}{K}$$

, and no greater than

$$n_2 = \frac{C_1 + (K-1)\sqrt{C_1 \log T}}{K}.$$

When $C_1 < t \leq C_1 + C_2$, denote $X^j$ as the number of rounds between the target arm get its $(j-1)^{th}$ and $j^{th}$ pick. After the target arm get its $(j-1)^{th}$ pick before its $j^{th}$ pick, in the worst case, its beta distribution is $B(j, 1+n_2)$, and that of any other arm is $B(1, 1+n_1)$. By simple arithmetic calculation, we have when $j = 1$, $\Pr\{\theta_1 < \theta_i\} = \frac{\beta}{1+\beta}$, and when $j \geq 2$, $\Pr\{\theta_1 < \theta_i\} \leq \frac{1}{j\beta}$ where $\beta = \frac{n_1+1}{n_2+1}$, so $\Pr\{\theta_1 > \theta_{i\neq1}\} \geq (1 - \frac{1}{j\beta})^{K-1}$. When $j = 1$, we have $\Pr\{\theta_1 > \theta_{i\neq1}\} \geq (\frac{\beta}{1+\beta})^{K-1}$. The probability that the target arm be selected is at least $1/2^{K-1}$, When $j < \frac{1}{\beta(1-2^{1-K})} := n_3$, and at least $1/2^{K-1}$. when $j \geq n_3$. With probability at least $1 - 1/T$, the target arm will be picked for at least $(C_2 - n_3(\frac{\beta}{1+\beta})^{1-K} \log T)/2 - \sqrt{C_2 \log T}$ rounds.

We select $C_1$ and $C_2$ to be large enough such that with high probability, when $t > C_1 + C_2$, $\theta_1 > \mu/2$ and $\theta_{i\neq1} < \mu/2$, so that the target arm will get all the picks. We set $C_1 = \frac{4 \log T}{\mu^2}$, and $C_2 = n_3(\frac{\beta}{1+\beta})^{1-K} \log T + 2\frac{\mu}{1-\mu}C_1$, then by $t = C_1 + C_2$, arms other than the target arm is picked for at least $n_1$ times, and the target arm's is picked for at least $n_2$ times with mean no less than $\mu$. By result from Agrawal and Goyal [2012], this can ensure that with probability at least $1 - K/T$, $\theta_{i\neq1}^t < \mu/2$ and $\theta_1^t > \mu/2$ true for all rounds. So with probability at least $1 - (2K+1)/T$, the target arm will get all picks when $t > C_1 + C_2$, with $C_1$ and $C_2$ as given above. $\square$

### A.5 Additional Experiments

Here we run both attack methods with or without knowing the mean reward $\mu$ of the target arm against UCB, Thompson sampling, and $\epsilon$-greedy bandit algorithms in different instances, where $\epsilon$ is set to be $T^{2/3}$ in $\epsilon$-greedy algorithm. For each pair of attack method and bandit algorithm, we run the experiments in three instances where there are two arms, and the mean reward for the optimal arm is always 1 while the mean reward for the target arm is $\mu = 0.3, 0.5, 0.7$ respectively. First we verify that our main attack algorithm 2 indeed manipulates the behavior of the bandit algorithms as the theory suggests. The parameters for this attack method is given by theorem 2 when attacking UCB algorithm, theorem 3 when attacking $\epsilon$-greedy algorithm, and theorem 4 when attacking Thompson sampling algorithm. In figure 3, for this attack method, we plot the number of rounds $n$ when the non-target arm get selected versus the total number of rounds $T$ for UCB algorithm in subfigure (a1),

Thompson sampling algorithm in subfigure (a2), and $\epsilon$-greedy algorithm in subfigure (a3). The plots show that there is a linear dependence between $n$ and $\log(T)$ in (a1) and (b1), and between $n$ and $T^{2/3}$ in (c1), which agrees with our theoretical guarantee. Each experiment is repeated for $100$ times.

Next we show that the modified attack which needs to estimate $\mu$ can also manipulate the algorithms without using a high corruption budget. In figure 4, in subfigures a1), b1) and c1), we plot the number of corruption rounds needed by the algorithm vs the total number of rounds $T$ in the case when the algorithm doesn't know the true mean $\mu$ for the bandit algorithms UCB, Thompson Sampling, and $\epsilon$-Greedy respectively. In subfigures a2), b2) and c2), we plot the corresponding number of times the non target arm was pulled for the corresponding corruption levels in the plots a1), b1) and c1) respectively. The plots show that even when the algorithm doesn't the mean reward, there is still a linear dependence between the corruption level $C$ and $\log(T)$ in (a1) and (b1), and between $C$ and $T^{2/3}$ in (c1), and similarly a linear dependence between the number of times the non-target arm is pulled $n$ and $\log(T)$ in (a2) and (b2), and between $n$ and $T^{2/3}$ in (c2). These results show that, along with strong theoretical guarantees, our attack methodologically also perform well empirically.

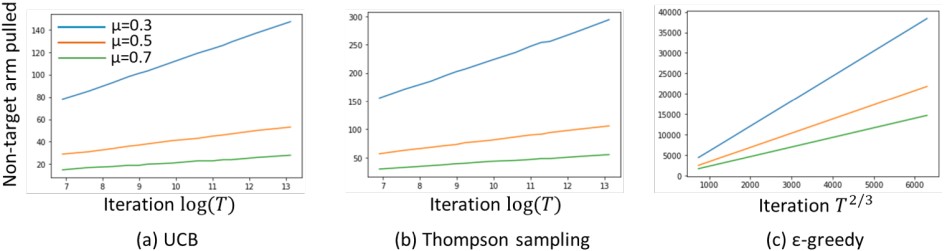

Figure 3: The attack which knows the mean reward of the target arm against (a) UCB algorithm, (b) Thompson sampling algorithm, and (c) $\epsilon$-greedy algorithm.

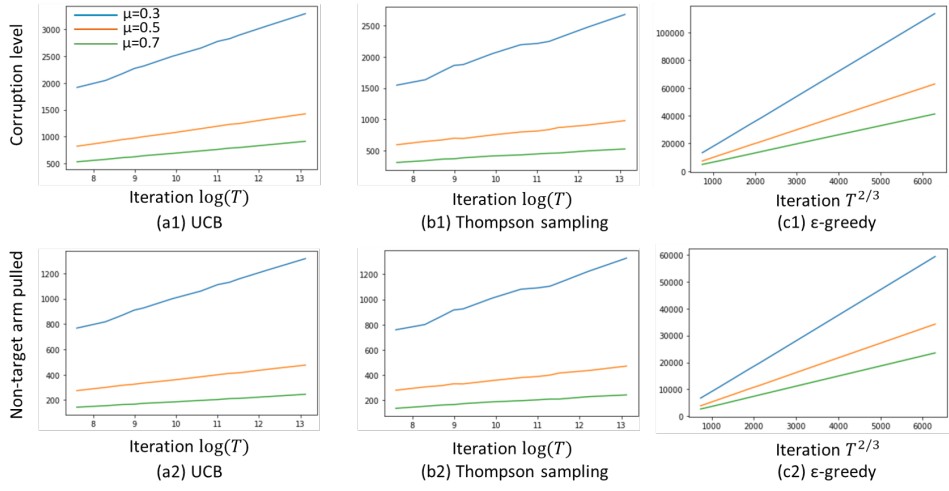

Figure 4: The modified attack which knows the mean reward of the target arm against (a1),(a2) UCB algorithm, (b1),(b2) Thompson sampling algorithm, and (c1),(c2) $\epsilon$-greedy algorithm.