# OpenReview forum: "Observation-Free Attacks on Stochastic Bandits"
_NeurIPS.cc/2021/Conference — NeurIPS 2021 Poster_

### Official Review · Reviewer_YeHZ · 2021-06-29

**Rating:** 6
**Confidence:** 5

**Summary:**

This paper proposes a corruption-based attacking framework for a general kind of MAB learning policies (which is called mean-based algorithms). They show that their framework can enforce all the mean-based algorithms to pull a target arm (which is not the optimal one) for $\Theta(T)$ number of times with only $o(T)$ number of corruptions. Then the authors consider three special (and common) mean-based algorithms, i.e., UCB policy, $\epsilon$-greedy policy and Thompson Sampling policy (with Beta prior). They show that under these learning policies, we can have better upper bounds for the needed corruption (to enforce the learning algorithm to pull the target arm for $\Theta(T)$ number of times). Specifically, $O(\log T)$ for UCB and Thompson Sampling, and $O(T^{2\over 3})$ for $\epsilon$-greedy. Finally, the authors run some experiments to demonstrate their theoretical results.

**Limitations And Societal Impact:**

Yes.

**Main Review:**

I really like the idea of the Observation-Free Attack framework in this paper, and I think this is the first result to use corruptions to attack Thompson Sampling policies. The definition of mean-based algorithms is very general and they are widely adopted in many works. Therefore, this paper is well-motivated and solves an important problem. I check most of the proofs and they seem to be correct. I also check some of the experiments, and the results accord with those in the paper.

Here are some of my questions:

1) In line 176, the authors claim that Thompson Sampling falls into the framework of mean-based algorithms. It is easy to check that Thompson Sampling (regardless of using Beta distribution or Gaussian distribution as the prior) follows the first two constraints in Definition 3. However, I find it not easy to prove that Thompson Sampling also follows the third constraint. Although I check the proof of Theorem 4 and it is correct, I think the authors may need to provide more details about why Thompson Sampling follows the third constraint in Definition 3.

2) The Observation-Free Attack framework require the knowledge of $T$. Can we release this assumption to design attacking method without knowledge about $T$? Usually we can adapt the doubling-trick for learning policies with requirement of known $T$, but it seems that the doubling-trick does not work for the attacking methods.

3) In the Observation-Free Attack framework, the learning agent may realize that there are corruptions. For example, when we are attacking the TS policy, after the first phase, the confidence interval of the expected reward of the target arm is a subset of $(0, \mu/2)$ (since the agent pull the target arm for about $\Theta(\log T/\mu^2)$) times. However, after the second phase, the confidence interval of the expected reward of the target arm becomes a subset of $(\mu/2, 1)$ (since we ensure that its sample value $\theta_i(t)$ is larger than $\mu/2$ with high probability). Then from this shift of the confidence interval, the learning agent can realize that his received reward is corrupted, and then maybe he can choose to start some defense mechanism. Therefore, can we improve the attacking method to make it impossible (or very hard) for the learning agent to realize corruptions?


==============After Rebuttal==============

Thanks for your replies. My score remains "6".

**Time Spent Reviewing:**

3

---

> ### Author Response · Authors · 2021-08-08
> **Response 3**
>
> 1. Indeed it is not easy to check that Thompson Sampling is a "mean based algorithm". We will add detailed explanation in the appendix.
> 2. It is indeed a limitation of the "observation free attack" that it requires the knowledge of $T$.
> 3. Great point! We also realize that agents can be aware of the attack by checking the shift of the confidence interval. It could be an additional constraint to the adversary that it cannot be detected by the agent. We think it is an interesting topic about adversarial attack to study in the next step.

---

### Official Review · Reviewer_jWoh · 2021-07-14

**Rating:** 5
**Confidence:** 4

**Summary:**

This paper designs a strategy to corrupt the rewards in stochastic bandits. In the setting, the adversary knows the true means of all arms before the interactions with the agent, and the corruption level --- the number of time steps that the adversary attacks any reward is bounded. It analyses how the mean-based algorithms will be impacted and presents the numerical performance of the attack strategy.

**Limitations And Societal Impact:**

As discussed above, the major limitations are:
1. Does the statement in Theorem 1 hold for all $K$-armed bandits or just 2-armed bandits? If it is only for $2$-armed bandits? Can it be generalized for any $K$?
2. I am not sure whether this work considers a weak adversary. The knowledge of all means of arms seems to be rarely assumed in existing works and I suspect it can let the adversary really 'powerful'. There seems to be a lack of motivation for this model.

**Main Review:**

This work studies an interesting question and is in general well-organized and easy to read. However, my major concerns are as follows:
1. The corruption strategy in the multi-armed bandits is an interesting topic, and the specific definitions and constraints vary among different papers. It would be important for the author(s) to provide more motivation for the specific setting considered in this paper. Why is this setting reasonable and interesting?
1. Regret minimization (RM) and best arm identification (BAI) are two major targets in the multi-armed bandits. While the author(s) introduced the concept of expected cumulative regret, which is the term to minimize for RM, the results in the coming theorems stating that the adversary can have the agent pull a certain arm (determined by the adversary) for a large fraction of times with high probability.\
   I am quite confused about which target does the adversary try to prevent the agent from achieving. RM, BAI, both, or some other target?\
   It would be better if this can be stated explicitly.
1. Though the adversary cannot observe the arm to pull before attacking the rewards, the adversary knows the true means of all arms in advance. Though the author(s) claims that the adversary is weak because of lack of observation,  I am concerned that the adversary may be too powerful. It would be better if there can be some explanation or motivation for this setting.
1. The corruption level is set to be the number of time steps the adversary attacks any reward, while in some earlier works it is defined as the summation of the corruptions added to all arms during the whole horizon.\
   In this sense, again the adversary seems to be powerful though he/she cannot observe the arm to pull. The corruption cost would be the same no matter he/she corrupts the reward of one or several arms at one time step.\
   It would be better if both definitions can be discussed, and there can be some discussion though the target of the agent may be different. Some numerical comparisons among different 'powers' of the adversary can be interesting.
1. Does the statement in Theorem 1 hold for all $K$-armed bandits or just $2$-armed bandits? \
   What is the related corruption level?
1. In Theorem 2-4, the result highly depends on the choices of $C_1$ and $C_2$. It would be clearer to write down all the $C_1$'s and $C_2$'s in the statements of the theorems.

Besides, there seems to be a minor typo in line 82 of page 2: 'to corruption every round' should be 'to corrupt every round'.



=========

Thanks for the detailed response from the author(s). Some of my concerns are solved. However, (i) I still doubt that if the corruption level is not important. The theorems do depend on the condition of the corruption level. (ii) As you say the target is RM, is it possible to write some corollaries stating what are the relevant regret bounds? I think this may make the contribution of the work clearer in some sense.

Anyway, I would like to increase my rating to 5 for the time being.

**Time Spent Reviewing:**

2.4

---

> ### Author Response · Authors · 2021-08-08
> **Response 2**
>
> 1. The most important part of the adversarial setting in this work is that the adversary cannot observe the action before attacking the rewards. Without such constraint, the adversary is too powerful such that no robust algorithm exists, and the papers which study robust algorithms always adopt such constraint. However, previous paper which study corruption strategies always lift this constraint, so they cannot be used to test the robustness of stochastic bandit algorithms. By studying corruption strategies under this constraint, we are more likely to discover the reasons why some algorithms are vulnerable, then one might design robust algorithms more easily in the future.
> 2. In this work, the target of agents is to minimize regret (RM). We call an algorithm vulnerable if it cannot guarantee sub-linear regret in expectation under data corruption attacks. When designing corruption strategies, the target of the adversary is usually harder than inducing linear regret. In this work, we adopt similar target for the adversary as previous papers, that is, maximizing the number of rounds when a target arm is pulled.
> 3. In appendix 6, we discuss the situation where the "observation free attack" doesn't know the mean rewards of arms in advance. It is true that when designing corruption strategies, the assumption that the attacker knows the mean rewards of arms is strong. We adopt this assumption only in the beginning to give concise result the proof, and we lift it eventually. Though when designing robust bandit algorithms, usually it is assumed that the attacker knows the mean rewards in advance, so that the robust algorithms in this sense are more reliable. Note that the assumption that the adversary is able to observe the action of the agent is too strong, since under this assumption, robust algorithm doesn't exist.
> 4. The definition of corruption level is not that important in this work. If the definition of corruption level is switched to the total amount of corruption, one can simply multiply the budget given in theorem 2-4 by a factor of the number of arms.
> 5. To show that an algorithm is vulnerable, one only need to show that the algorithm will have linear regret in a specific case with an adversary of sub-linear corruption level. The specific cases we used here are $2$ - armed bandit. We show that the any "mean based" algorithm will suffer from linear regret in at least one of these cases given sub-linear corruption level, so all of them are vulnerable.
> 6. Thanks for the advice, we will add $C_1$'s and $C_2$'s in the statements of theorems.

---

> > ### Comment · Reviewer_jWoh · 2021-09-01
> > **Comment after rebuttal**
> >
> > Hi,
> >
> > Thanks for your response. Some of my concerns are solved. However,
> > (i) I still doubt that if the corruption level is not important. The theorems do depend on the condition of the corruption level.
> > (ii) As you say the target is RM, is it possible to write some corollaries stating what are the relevant regret bounds? I think this may make the contribution of the work clearer in some sense.
> >
> > I would like to increase my rating to 5.

---

### Official Review · Reviewer_H7xu · 2021-07-16

**Rating:** 4
**Confidence:** 4

**Summary:**

This paper studies adversarial attacks against no-regret stochastic bandit algorithms, including UCB, epsilon-greedy and the Thompson sampling algorithms. The attacker has the ability to perturb the reward function, but such perturbation must happen before observing the action selected by the bandit player. The paper developed a three-phase attack strategy, where in the first phase, the attacker sets the reward to 0 for every arm. As a result, every arm can be selected often enough and the confidence intervals of all arms are small, while the empirical mean are all 0 as well. Then in the second phase, the attacker boosts the reward for the target arm to the maximum, and the authors show that during the second phase, the target arm will always be selected. Finally, in the third phase, no attack happens, and due to the increase of reward for the target arm in phase two, the authors proved that in the third phase, the target arm still maintains its superiority and the other arms cannot recover. Therefore, the target arm will be pulled in almost every round with high probability. The empirical results in this paper demonstrated the effectiveness of the proposed attack algorithm.

**Limitations And Societal Impact:**

Please discuss potential negative implications of the theoretical results developed in this paper. Furthermore, what are some potential countermeasures to mitigate the effect of adversarial attacks to make modern bandit-based decision systems more robust.

**Main Review:**

I have several questions and concerns about the results presented in the paper.

First, I am not sure if the proof is correct. Specifically, in the proof for Theorem 2 (the first equation after line 522), it’s not clear why UCB1^t>=sqrt(logT/(C1/K)) holds. I hope the authors could provide a more detailed proof on this inequality. Second, the proof requires choosing C1>=KlogT/mu1, where mu1 is the true mean of the target arm. What if mu1=0? Then no such C1 can be selected. I feel like a lot of places in the proof are not rigorous enough.

Second, the following paper (which is cited in the paper) also studied an attacker who perturbs the reward function before observing the selected arm. However, it is proved that the regret only scales linearly as the total corruption budget C grows. In the case that C=O(log T), the regret also grows logarithmically, which is in odd with the conclusion of this paper. I wonder if the authors can explain why these two results are contradicting each other?

“Stochastic bandits robust to adversarial corruptions”

Again in Theorem 3, when the mean reward mu of the target arm is 0, the result breaks. I wonder how to fix this theoretical caveat?

In this paper, the authors made the assumption that the reward function after corruption is bounded. Furthermore, since the attacker designs the post-attack reward function before observing the selected action, I feel like the setting becomes equivalent to the adversarial bandit problem. Then I wonder could we just apply standard adversarial bandit algorithms like EXP3 to defend against this attack? In doing so, at least we can guarantee sublinear regret, which seems to suggest that the target arm cannot be pulled in almost every rounds. Otherwise, the regret would be linear. I wonder if the authors could provide a more comprehensive and deeper discussion on how the theoretical results in this paper connects to adversarial bandits.

In summary, my concern about the paper is on the theory. Some places do not seem absolutely correct. Moreover, intuitively the results are not perfectly aligned with some prior works and the adversarial bandit. Therefore, I hope that the authors can explain those points in detail in the response. Thanks.


**Time Spent Reviewing:**

5

---

> ### Author Response · Authors · 2021-08-08
> **Response 1**
>
> 1. The inequality in line 522 is given by the fact that the derivative over $n$ of the left-hand side of the inequality is negative. We will explain this in more detail in the appendix.
> 2. Theorem 2 claims that all 'mean based bandit algorithms', rather than any bandit algorithms, are vulnerable to adversarial data corruption attack. The algorithm given by “Stochastic bandits robust to adversarial corruptions” is not a "mean based bandit algorithm"
> 3. Theorem 3,4,5 shows the budget that "observation-free attack" needs to manipulate several classical bandit algorithms. So indeed when the mean reward of the target arm is or close to 0, the budget that "observation-free attack" needs can exceed $T$, which means this attack is unable to manipulate these bandit algorithms in such cases.
>
> The reviewer probably misunderstand that the "observation-free" attack in this work can efficiently attack any bandit algorithms, which is not true. This work studies what makes a bandit algorithm vulnerable to data corruption attack, and one answer this word finds is the ignorance of bias in estimation induced by the adversary, which the robust algorithms don't do. The robust algorithms found by previous works, including EXP3, does not belong to the class of vulnerable algorithms in this work. This work aligns with the previous work on adversarial bandit.

---

### Decision · Program_Chairs · 2021-09-27

**Decision:**

Accept (Poster)

**Comment:**

This paper is the first one to provide an observation-free attack against some of the well-known bandit algorithms. While there are many existing work on adversarial attacks against bandit algorithms, those attacks assume stronger attacker model which can observe the behaviour of the algorithm and then adapt the future attacks accordingly. This paper relaxes this attacker model by not allowing this observation and thus, making the attack much harder.  The main research question is then whether all the bandit algorithms can be attacked successfully under this attack scheme. The paper provides a partial answer to this question by showing that mean-based bandit algorithms such as UCB and TS are still vulnerable against this attack.

As someone quite familiar with the topic of adversarial attacks on bandits, I find this paper to be very interesting. In fact, it introduces a 3rd attacker model (the other 2 are strong attackers - where the manipulation happens after observing the action of the algorithm, and weak attacker model - where the attack has to happen before the action, but the action is still observed). By introducing this new model, the authors managed to show that there are indeed differences in the behaviour of different bandit algorithms. In fact, some algorithms are vulnerable against this attack, while other are not. Note that with the other 2 attacker model all the algorithms would be vulnerable in the same way. This finding provides new insights to the way bandit algorithms work.

In more detail, for the other 2 (existing) attacker models, there are universal and successful attack strategies. Thus, in those attacker models it's hard to really understand whether there are differences between the behaviour of each bandit algorithm against the attack. On the other hand, in this new attacker model of this paper, there is a clear difference (i.e., mean-based vs. non mean-based). I believe this new insight makes the paper's contribution even more interesting. As such, I believe this work is quite interesting and is worth being presented at the conference.

Regarding the comments and criticisms from the other reviewers: I don't agree with the comments regarding the technical issues. I went through the proofs myself and I found the authors' responses to be helpful and clarified the concerns. I also disagree that this model is very similar to the adversarial bandit setting, as the underlying regret notion is not exactly the same (in adversarial bandits we measure the regret based on the observed feedback, assuming that they are correct), while in attacks on bandits problems the regret is measured on the true (and unobservable) rewards. it is true that if the contamination budget is small (e.g., O(logT)) then Exp3 and other adversarial bandit algorithms would recover well. But again, they are not mean-based methods.

I have one criticism though: From the authors' responses and comments, it seems that it is not too difficult to show that non mean-based methods such as Exp3 are still robust against this type of attacker model. If the authors can add this to the paper, that would make the story complete. Therefore my recommendation of acceptance would rely on the assumption that this argument will be added to the final version of the paper.